# Factors Influencing Breast Milk Fat Loss during Administration in the Neonatal Intensive Care Unit

**DOI:** 10.3390/nu13061939

**Published:** 2021-06-05

**Authors:** Mattias Paulsson, Lena Jacobsson, Fredrik Ahlsson

**Affiliations:** Department of Children’s and Women’s Health, Uppsala University Hospital, SE 751 85 Uppsala, Sweden; lena.jacobsson@regiongavleborg.se (L.J.); fredrik.ahlsson@kbh.uu.se (F.A.)

**Keywords:** bolus feeding, breast milk, continuous feeding, fat loss, tube feeding

## Abstract

The objective of this study was to investigate factors influencing fat loss during tube feeding of breast milk to preterm infants. An experimental study with 81 feeding simulations was performed, with nine continuous infusions in each of six modalities: Horizontal Higher, Horizontal Matched, Horizontal Lower, Tilted Higher, Tilted Matched, and Tilted Lower, and for comparison, 27 bolus feedings: nine flushed with air, nine with water, and nine that were not flushed, done at matched height. Each simulation utilized 16 mL of breast milk given over four hours. Continuous infusions were given with a flow rate of 4 mL/h. Bolus was given as 8 mL over the course of 15–20 min every other hour. Analysis for fat, true protein, carbohydrate, total solids, and energy was performed before and after each simulation. The percent of macronutrient loss was compared between all simulations. Continuous infusion resulted in an average fat loss of 40%. Bolus feedings resulted in an average fat loss of 11% (*p* ≤ 0.001). Considerable fat loss is seen during continuous tube feeding. Neither height in relation to the infant nor tilting of the pump reduce fat loss. To limit fat loss, the bolus feeding method should be utilized.

## 1. Introduction

Extrauterine growth restriction (EUGR) is common among preterm infants [1] and very low birth weight (VLBW) infants [2]. One study identified an incidence of EUGR of 73.3% in VLBW infants [3]. While the cause of EUGR is multifactorial, it is well agreed that nutritional support is a critical factor [4,5]. Infants cared for in neonatal intensive care units (NICUs) with nutritional support teams or where early aggressive nutrition is used have a lower incidence of EUGR than those infants cared for in NICUs that do not have this [3,6].

Enteral nutrition is provided by tube feeding until premature infants overcome the challenges of breast or bottle-feeding. The method, however, is not without difficulties, and it is a contributor to pre-exposure fat loss [7]. Lipid losses during tube feeding were proved already in 1978 by Brooke et al. [8]. Their study found that the energy content loss, representing fat loss, was much higher at the end of a feeding as compared with at the beginning, with variations in energy of up to 24%.

Narayanan et al. demonstrated that tilting the feeding pump at an angle between 25 and 40 degrees with an eccentric nozzle is the most successful method of reducing lipid losses, with a fat loss of as little as 7.1% occurring with the pump tilted [9]. The slower the flow rate of a continuous feeding, the higher the fat loss, indicated by the results of Stocks et al. [10], and effects after heating, freezing, refrigeration, and tube feeding, discussed by Tacken et al. [11].

It is well established that bolus feedings yield smaller fat losses than continuous feedings [8,9,10,12,13,14]; nevertheless, continuous feedings are still used in NICUs. Continuous feeding seems to be better than intermittent feeding with regard to gastrointestinal tolerance in VLBW [15]; bolus-fed infants had a significantly higher risk of behavioral stress response [16], and continuous feeding has been shown to lead to infants reaching total enteral feeding faster and significantly faster lower leg growth rate as compared with bolus-fed infants [15]. However, another study reports a similar growth rate when fed with continuous infusion compared with bolus feed [17].

With the benefits of continuous feeding in mind, our aim is to find factors influencing fat loss in feeding methods and be able to give clinically feasible recommendations. Our hypothesis is that the position of the breast milk syringe in relation to the infant’s position can influence the fat loss due to the inhomogeneity of the breast milk emulsion, also known as creaming (migration of lipid droplets under the influence of buoyancy).

## 2. Materials and Methods

This was an experimental study utilizing surplus donated human breast milk (BM) from the neonatal intensive care unit at Uppsala University Children’s Hospital, Sweden. Because no patient details were handled and the milk was not required in the clinic, no application for ethical vetting was needed.

The milk was collected between June and October 2017 and kept in the freezer according to the normal clinical routine of the milk bank at Uppsala University Children’s Hospital. All milk utilized was from mothers of prematurely born infants, but no record of infant postmenstrual age at the time of collection was kept. The milk was donated for clinical use and made available for this study on the expiry date of clinical use (six months shelf life in freezer). Experiments were executed between January and October 2018. The evening prior to testing, the milk was moved to the refrigerator for thawing, alternatively placed in a water bath of 35–40 degrees Celsius on the day of testing. As per local regulations, thawed BM was kept for a maximum of 24 h in the refrigerator before analysis.

### 2.1. General Simulation Setup

The BM administration simulation was performed using the standard NICU tube feeding system (Carefusion, Alaris Enteral Syringe Pump, Basingstoke, UK), an infusion syringe (VYGON, Nutrisafe2, 60 mL, ref 1015.603, Vygon, Ecouen, France) connected to an extension set (VYGON, Nutrisafe2, Internal diameter: 1.5 mm, External diameter 2.5 mm, Length 150 cm, ref 368.152), and lastly a feeding tube (VYGON, Nutrisafe2 (PUR), 06Fr, Length 50 cm, polyurethane, ref 1361.062). The syringes were placed in the pump with the eccentric nozzle at 12 o’clock. The BM was collected into 30 mL plastic medicine cups (Hammarplast Medical AB, ref 10300) covered with laboratory parafilm (Parafilm M, Bemis/Amcor, Zürich, Switzerland) to simulate an infant’s stomach. A map pin was used to poke a hole in the parafilm to insert the feeding tube into the medicine cup. The medicine cup was then placed 100 cm from the discharge orifice of the syringe in order to mimic placement in the wards. Breast milk samples were measured as pairs (before/after) to estimate the effects of handling relative to the starting sample. Experiments (both continuous described in Section 2.2 and bolus in Section 2.3) were run at room temperature for 4 h in order to standardize the effects of, e.g. enzyme activity on content.

### 2.2. Continuous Feeding Setup

The milk was administered through six different continuous modalities: Horizontal Higher (HH), Horizontal Matched (HM), Horizontal Lower (HL), Tilted Higher (TH), Tilted Matched (TM), and Tilted Lower (TL). Horizontal was defined as the syringe being parallel to the working table surface. Tilted was defined as the syringe pump being tilted 40–45 degrees with the discharge orifice placed in an upward slant (Figure 1). Matched was defined as the orifice being placed at the midpoint of the plastic medicine cup, or “infant”. Higher was defined as the orifice being 30 cm above the midpoint of the “infant” and Lower was defined as the orifice being 30 cm lower.

The NICU 95F only had one 45-degree pole clamp. Two more pole clamps were constructed for this study (Figure 2).

Each modality was tested a total of nine times with a continuous administration of BM at a flow rate of 4 mL/h for approximately four hours. The flow rate was calculated in accordance with local clinical protocol based on a 600 g infant (recommended daily intake is 150–170 mL/kg/day). The duration of four hours per test run was used based on local clinical protocol that feeding syringes are to be changed every four hours. To deliver 16 mL, there is a need for some overage. It was noted that, repeatedly, 2–3 mL was left in the syringe with a visibly high fat content, possibly as a strategic limitation of the pump to avoid delivering air in the tubing and true to the clinical setting. Of the 16 mL initially aspirated into the syringe, about 3 mL is needed to fill the extension tubing. Normal clinical practice dictates that the syringe be filled with an extra 3 mL for the first feeding to fill the extension tubing and ensure the right amount of BM is delivered. This was not done, as the BM supply was limited.

### 2.3. Bolus Feeding Setup

Twenty-seven bolus feedings were also simulated. Bolus was administered through the same 16 mL syringe coupled directly to the feeding tube and then collected in 30 mL medicine cups. Bolus was defined as manually giving 8 mL of BM over the course of 15–20 min every other hour. This was done twice to equal 16 mL per four hours. Afterwards the feeding tube was flushed with 1 mL, equal to the dead space of the feeding tube, of either water or air (nine simulations of each) as per local clinical practice. An additional nine bolus feedings were simulated without flushing. After the second administration of 8 mL BM and flushing of either air or water, 10 mL was collected from the medicine cup and poured into a test tube for analysis. It should be noted that the bolus feedings that were not flushed were by mistake connected to an extension set (described in Section 2.1) before being coupled to the feeding tube. The percent of fat loss was a little greater for these feedings, but still within the margin of error.

### 2.4. Analyses

After each test run, 10 mL of BM was poured from the medicine cup into a test tube and then analyzed for the following: fat, true protein, carbohydrate, total solids, and energy content using the mid-infrared technology Miris Human Milk Analyzer (Miris HMA™ Uppsala, Sweden). The test tubes of 10 mL of BM were, for the first part of the study (January–May), placed in a water bath (stainless steel beaker filled with warm water) with a temperature of 35–40 degrees Celsius (determined by glass thermometer) or, for the last part (from September 2018), a bead bath (Miris Heater™) set at 40 degrees Celsius for a minimum of five to ten minutes. The BM was then homogenized using Miris SONICATOR™ for a total of 15 s as per manual instructions. During the study, two different, but equivalent sonicators were used as the first broke in February 2018 and the replacement sonicator was used from May 2018 (n = 60).

Analysis started after homogenization, with three separate tests of 3 mL of BM being analyzed and then the average of each parameter was calculated.

The data were analyzed using IBM SPSS Statistics. A one-way analysis of variance (ANOVA) with post-hoc Tukey test was performed to determine statistically significant differences between modalities, with statistical significance being reached if *p* < 0.05. The average mass concentration (including standard deviation) of each macronutrient prior to the feeding simulation was calculated and subsequently compared to the average mass concentration (including standard deviation) after the feeding simulation. The results from continuous feeding measurements were compared with bolus feedings by calculating the difference between the average mass concentrations of each macronutrient. All continuous feeding simulations were merged and compared with the merged bolus feeding simulations.

## 3. Results

The average fat concentration before the continuous feeding simulations was between 2.3 and 3.3 g/100 mL compared with the average fat concentration afterwards of between 1.2 and 2.0 g/100 mL (Table 1).

The bolus feeding simulations had an average fat concentration before the feeding simulations of 2.5 g/100 mL, 2.5 g/100 mL, and 3.5 g/100 mL for bolus with air flush, bolus with water flush, and bolus with no flush, respectively (Table 1). The average fat concentration after the bolus feeding simulations was 2.3 g/100 mL when flushed with air, 2.2 g/100 mL when flushed with water, and 3.1 g/100 mL with no flush.

Energy content was an average of 62–67 kcal/100 mL before as compared with 53–56 kcal/100 mL after for the continuous feeding simulations. The bolus feeding simulations had a range of 65–73 kcal/100 mL before as compared with an average content of 65 kcal/100 mL, specifically, 64 kcal/100 mL, 61 kcal/100 mL, and 70 kcal/100 mL for air flush, water flush, and no flush, respectively.

An average fat loss between 38% and 45% occurred when using the continuous feeding method; meanwhile, the bolus feeding method only resulted in an average of 8%, 12%, and 12%, respectively (Table 2). An average energy loss of 13.6% occurred during continuous feedings and 4.9% during bolus feedings, 2.3%, 7.2%, and 4.7%, respectively (Table 2).

There was statistically significant fat loss after continuous tube feeding compared with the bolus feedings, as determined by one-way ANOVA (F (6, 56) = 7,330, *p* = 0.000) with post-hoc Tukey test. However, no significant difference in fat loss was seen between the horizontal and tilted modalities (*p* = 0.875–1.000 depending on modality). Neither was there a significant difference in fat loss between the three different bolus-feeding methods.

Visually, less fat accumulation was seen in the syringe during tilted feeding simulations compared with horizontal (Figure 3). This was not significant in actual calculations of fat content.

## 4. Discussion

A significant decrease of fat content was seen between the continuous feeding simulations and the bolus feeding simulations, as expected. However, the fact that no significant difference was shown between the horizontal and tilted modalities, as well as height placement of the pump in relation to the infant, disproved the study’s hypothesis. These findings directly oppose those of Narayanan et al. [9]. Our hypothesis was based on the fact that, when the breast milk fat emulsion separates (creaming), the fat rises owing to density differences, meaning that, within the syringe in a pump placed lower than the infant, the fat rises towards the infant. Unfortunately, the length and size of the extension tubing the milk must travel through used in today’s neonatal care seem to render the process of creaming useless in reducing fat loss. Ultimately, this leads to the conclusion that neither the pumps placement in relation to the infant nor tilting of the pump has a role in reducing fat loss and that, as of today, the only way to reduce fat loss is to utilize bolus feeding instead of continuous.

As previously stated, our findings are in contrast to the findings of Narayanan et al., which showed a significant decrease in fat loss when using a feeding pump tilted between 25 to 40 degrees [9]. With this study being done 34 years after the previously mentioned study, newer technology, newer enteral feeding systems using other plastic materials, and newer methods for analyzing human breast milk may have yielded these opposing results. Moreover, Narayanan et al. do not specify in their study method what flow rate was used or the size or length of the tubing and syringe, which would also affect the results. Perhaps, tilting of the pump is meaningful when using other flow rates than those we tested, rendering the comparison of the two studies impractical as our study and flow rates pertain specifically to preterm and VLBW infants. Zozaya et al. showed that the amount of lipids adsorbed to tube surfaces is low, only 0.6%, for 24 h infusion, but higher (13%) in a simulated 30 min infusion [18]. The higher amounts of fat loss seen in this and other studies are likely caused by creaming in combination with lipid being trapped in tube connections and syringe in built upon material adsorption processes.

Higher flow rates have been shown to result in less fat loss [10]. Our study supports these findings because, with a flow rate of 4 mL/h, we observed an average fat loss of 40%, which is more than the 24% lost according to the study of Brooke et al., with a flow rate of 10–25 mL/h [8]. Bolus feedings are given at a much higher flow rate than continuous, with ours given at an estimated 20–30 mL/h. This is important to note as our continuous flow rates reflect those that are used for VLBW infants and preterm newborns, meaning that those infants are possibly at a higher risk of malnutrition and EUGR when using the continuous feeding method.

Discussing the methodologies used, there are more accurate methods available to analyze fat content in milk than the chosen mid-IR transmission spectroscopy. However mid-IR transmission spectroscopy is being used in the NICUs around the world [19] and clinicians rely on the results for optimizing nutrition for extremely preterm infants. In addition, results and conclusions are handled on relative calculations of paired samples rather than absolute quantifications. Homogenization using ultrasound prior to tube feeding is shown not to alter the lipid composition nor cause significant difference in fat loss between ultrasound treated milk and non-treated milk [19]. It was not possible, in this study setup, to specify the nutrient content variation based on the postmenstrual age at breastmilk collection. Preferably, after clinical translation of our findings, performing analysis of macronutrients in the milk before feeding can be used to acknowledge this aspect.

Although this study is conducted in an experimental setting, we have used the same equipment as is used in our NICU. Thus, the discovered fat loss mirrors the fat loss when using continuous tube feeding in the clinical setting and the findings are easily translated into clinical use.

Brooke et al. and Narayanan et al. both concluded that regular agitation of the syringe, hourly and half-hourly, respectively, was a possible solution to the problem of fat loss during continuous feedings [8,9]. García came to the same conclusion; however, we are in agreement with the numerous before us who have argued that this method is time consuming and thus not advantageous [20]. This again leads to the deduction that bolus feedings are superior in preventing fat loss when tube feeding the preterm infant. If fat losses during tube feeding can be reduced, the infant is also likely to receive a higher amount of fat-soluble vitamins.

To conclude, this study could not show that position nor tilting angle of pump were factors influencing fat loss. Accordingly, the only clinically feasible recommendation from this study is to use bolus feed to avoid the significant fat loss otherwise seen in continuous tube feeding. Further research of interest would be to study the effects of shorter feeding tubes as well as the sorption phenomena occurring within the tubing by comparing tubing materials with different surface properties.

## Figures and Tables

**Figure 1 nutrients-13-01939-f001:**
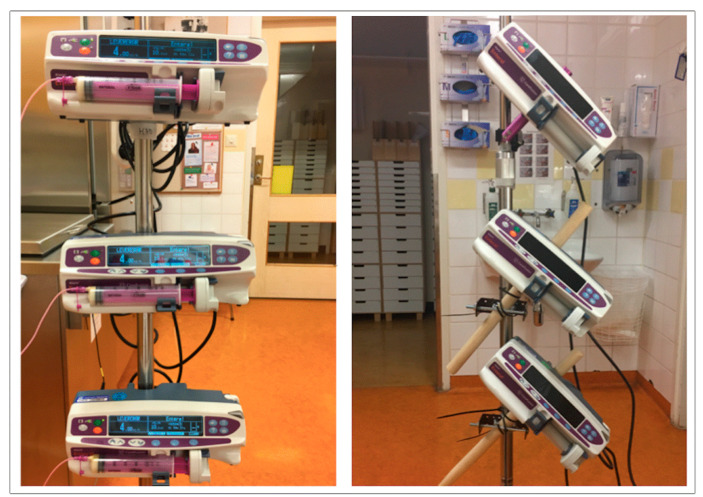
Horizontal and Tilted Pump Set Up. Photo by author.

**Figure 2 nutrients-13-01939-f002:**
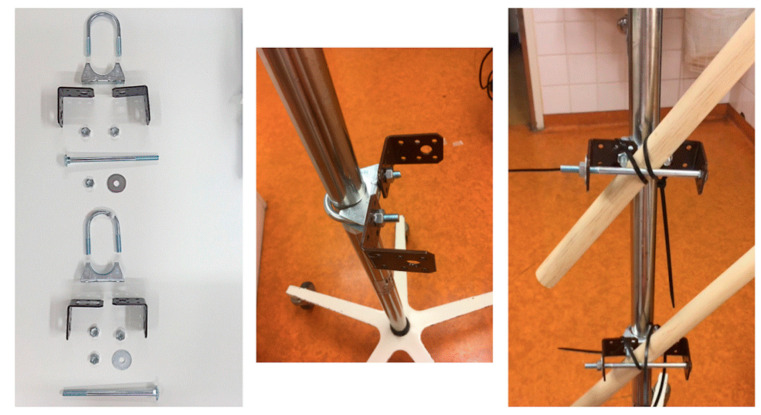
Construction of Tilted modality. Photo by author.

**Figure 3 nutrients-13-01939-f003:**
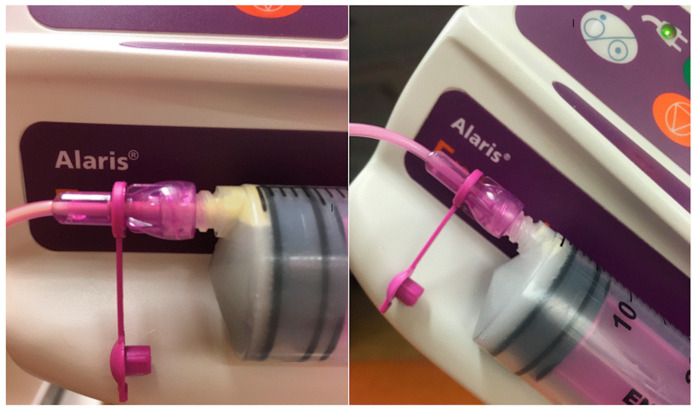
Visual fat loss after tilted and horizontal feeding simulations. Photo by author.

**Table 1 nutrients-13-01939-t001:** Average mass concentration of macronutrients before and after feeding simulations (n = 9).

	Fat				Carbohydrate				Total Solids				Energy				Total Protein			
	Before		After		Before		After		Before		After		Before		After		Before		After	
	Mean	SD	Mean	SD	Mean	SD	Mean	SD	Mean	SD	Mean	SD	Mean	SD	Mean	SD	Mean	SD	Mean	SD
Horiz. Higher	3.04	0.97	1.79	0.43	7.02	1.02	7.14	0.94	12.31	1.10	11.21	0.86	65.33	8.02	54.33	3.97	1.67	0.32	1.69	0.30
Horiz. Matched	3.08	0.96	1.83	0.42	6.93	0.93	7.13	0.87	12.30	1.09	11.33	0.85	65.44	8.02	54.89	4.14	1.70	0.30	1.77	0.25
Horiz. Lower	3.26	0.95	1.97	0.80	6.96	1.01	7.06	1.00	12.50	1.16	11.44	0.90	67.22	8.33	56.11	6.11	1.71	0.35	1.80	0.31
Tilted Higher	2.42	0.70	1.46	0.42	7.73	0.53	7.71	0.54	12.62	0.47	11.72	0.37	63.56	5.55	54.67	3.32	1.87	0.40	1.89	0.43
Tilted Matched	2.27	0.40	1.22	0.21	7.89	0.45	7.89	0.45	12.53	0.32	11.64	0.19	62.33	3.16	53.00	1.80	1.79	0.42	1.83	0.44
Tilted Lower	2.42	0.70	1.41	0.36	7.73	0.53	7.73	0.52	12.62	0.47	11.76	0.21	63.56	5.55	54.56	2.19	1.87	0.40	1.92	0.40
Bolus Air	2.50	0.00	2.30	0.05	8.40	0.00	8.46	0.05	13.10	0.00	13.00	0.07	65.00	0.00	64.00	0.50	1.60	0.00	1.60	0.00
Bolus Water	2.50	0.00	2.20	0.00	8.40	0.00	7.97	0.05	13.10	0.00	12.28	0.07	65.00	0.00	60.78	0.44	1.60	0.00	1.50	0.00
Bolus No Flush	3.51	1.07	3.10	0.99	7.12	0.59	7.10	0.58	13.63	1.10	13.27	1.07	73.33	9.89	69.78	9.51	2.31	0.55	2.31	0.55

Mass concentration presented in g/100 mL. Energy presented in kcal/100 mL. Mean and Standard Deviation (SD).

**Table 2 nutrients-13-01939-t002:** Mean percentage (%) difference of mean mass concentration of macronutrients. Part A: After feeding simulations; Part B: Summary after continuous and bolus feeding simulations.

**Part A** **(n = 9 for Each Row)**	**Fat**	**Carbohydrate**	**Total Solids**	**Energy**	**True Protein**
Horizontal Higher	−38.00	2.22	−8.56	−15.67	0.89
Horizontal Matched	−37.78	2.11	−5.44	−9.22	0.11
Horizontal Lower	−38.56	2.11	−7.78	−15.00	4.44
Tilted Higher	−39.11	−0.11	−6.89	−13.56	1.22
Tilted Matched	−44.78	0.22	−7.11	−14.44	3.22
Tilted Lower	−41.11	0.22	−6.89	−13.67	3.33
Bolus Air	−8.00	0.11	−.89	−2.33	2.44
Bolus Water	−12.00	−5.33	−6.33	−7.22	−4.00
Bolus No Flush	−12.00	−0.22	−2.78	−4.67	0.33
**Part B**	**Fat**	**Carbohydrate**	**Total Solids**	**Energy**	**True Protein**
Continuous, n = 54	−39.89	1.17	−7.10	−13.59	2.20
Bolus, n = 27	−10.80	−1.99	−3.50	−4.91	–2.04

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
