# Peer review of "Factors Influencing Breast Milk Fat Loss during Administration in the Neonatal Intensive Care Unit"

_nutrients, 2021, doi:10.3390/nu13061939_

Round 1
Reviewer 1 Report
Important clinical question as optimising nutrition in preterm neonates is a priority.
Agree that this was an experimental simulated design but authors will have to include a paragraph in discussion on translational relevance in the clinical context
Also authors write that they collected milk in 2017 and conducted the experiment in 2018. As per my reading, PDHM starts losing fat content (due to action of internal lipases) and cannot be stored beyond 3 months in the best storage conditions, please comment on this aspect.
Also acknowledging that postmenstrual age at collection of the milk samples was unknown is important as these factors would be taken into consideration for a clinical study.
Author Response
Reviewer 1: Important clinical question as optimising nutrition in preterm neonates is a priority.
We thank Reviewer 1 for the time invested in the peer-review of our manuscript, and for the constructive feedback provided.
Agree that this was an experimental simulated design but authors will have to include a paragraph in discussion on translational relevance in the clinical context
Thank you for this comment. We have added the following sentence in the discussion: “Although this study is conducted in an experimental setting, we have used the same equipment as is used in our NICU. Thus, the discovered fat loss mirrors the fat loss when using continuous tube feeding in the clinical setting and the findings are easily translated for clinical use.”
Also authors write that they collected milk in 2017 and conducted the experiment in 2018. As per my reading, PDHM starts losing fat content (due to action of internal lipases) and cannot be stored beyond 3 months in the best storage conditions, please comment on this aspect.
We thank the Reviewer for this question. When working with breastmilk we were obliged to use breastmilk on the expire date since it would have been unethical to use breastmilk that an extremely preterm infant could had benefited from. Further, we used a before and after approach when conducting the experiment. The breastmilk was thawed and before it was administered a sample of the milk was take and analysed in the MIRIS. Than from the same batch the milk was administered according to protocol. Thus, the breastmilk served as its on control as is described in Methods section 2.1. We have also added (row 65) the following information “(six months shelf life in freezer)”.
Also acknowledging that postmenstrual age at collection of the milk samples was unknown is important as these factors would be taken into consideration for a clinical study.
Thank you for pointing this out. - We fully agree it is of importance since the breast milk alter its composition depending of the postmenstrual age. We have added the following sentence in the section discussing translational relevance: ”It was not possible, in this study setup, to specify the nutrient content variation based on the postmenstrual age at breastmilk collection. Preferably, after clinical translation of our findings, performing analysis of macronutrients in the milk before feeding can be used to acknowledge this aspect.”
Reviewer 2 Report
This is an interesting and important study of the effects of syringe orientation and flow rate upon nutrient loss during infusion of breastmilk through enteral feeding tubes. The authors found that changing the orientation of the syringe had no effect upon the loss of lipid from the milk delivered. They did find a significant difference between bolus and continuous feeding modes. The study is well written and the data is carefully collected and analysed and clearly presented.
Two minor quibbles: The authors did not extend their study to include analysis of fortified breastmilk as well. However, I understand that the Miris HBM analyser isn’t validated on such material and so this would complicate the study by necessitating more complex analytical methods.
Secondly, as breastmilk is a ‘live’ secretion, enzyme activity can change its constituency over time, especially at room temperature. As the continuous feeding mode took place over a four hour period how did the authors deal with this potentially confounding effect upon their measurements?
A few corrections and clarifications:
Line 39 states that energy content was higher at the end of feeding. I think this is a mistake. Please clarify.
Line 124: how long was this extension?
Lines 144-150 are confusing and need to be edited for clarity.
Line 175: It is not clear what treatments are being compared in the statistical analysis. This requires elaboration to explain where the significance lies. Is the difference between original milk and bolus feeding significant? And what about original milk and continuous feeding? For each significant test the results of the ANOVA (the F statistic, the degrees of freedom and the thresholds exceeded by the p value) should be presented. This data could be added to a table if it is extensive or, if there are only a few results to present, they can be embedded in the text.
Line 224: the claim that “mid-IR transmission spectroscopy is being used in the NICUs around the world” needs a reference to support it.
Line 225: really is a typo of rely
Author Response
Reviewer 2: This is an interesting and important study of the effects of syringe orientation and flow rate upon nutrient loss during infusion of breastmilk through enteral feeding tubes. The authors found that changing the orientation of the syringe had no effect upon the loss of lipid from the milk delivered. They did find a significant difference between bolus and continuous feeding modes. The study is well written and the data is carefully collected and analysed and clearly presented.
We thank Reviewer 2 for the time invested in the peer-review of our manuscript, and for the constructive feedback provided.
Two minor quibbles: The authors did not extend their study to include analysis of fortified breastmilk as well. However, I understand that the Miris HBM analyser isn’t validated on such material and so this would complicate the study by necessitating more complex analytical methods.
We agree with Reviewer 2 that the effects on fortified milk is also an interesting field of research and that including e.g. GC-MS technique for analysis is a highly relevant approach for further research.
Secondly, as breastmilk is a ‘live’ secretion, enzyme activity can change its constituency over time, especially at room temperature. As the continuous feeding mode took place over a four hour period how did the authors deal with this potentially confounding effect upon their measurements?
We agree with Reviewer 2 that eg lipase activity can change the constituency over time so this is important to consider. In order to reduce this potential confounder, the experimental setup had a fixed time frame of 4 hours both for the bolus feed and the continuous infusion. Row 82 in the methods section has been updated with the text “Experiments (both continuous described in section 2.2 and bolus in 2.3) were run at room temperature for 4 h in order to standardize the effects on e.g enzyme activity on content.”
A few corrections and clarifications:
Line 39 states that energy content was higher at the end of feeding. I think this is a mistake. Please clarify.
We thank Reviewer 2 for pointing this incorrect reference aim. The text was missing the word “loss” and now updated.
Line 124: how long was this extension?
This was the standard extension set (described in 2.1) and the text on line 124 has been updated accordingly.
Lines 144-150 are confusing and need to be edited for clarity.
We thank Reviewer 2 for highlighting this. The text has been edited.
Line 175: It is not clear what treatments are being compared in the statistical analysis. This requires elaboration to explain where the significance lies. Is the difference between original milk and bolus feeding significant? And what about original milk and continuous feeding? For each significant test the results of the ANOVA (the F statistic, the degrees of freedom and the thresholds exceeded by the p value) should be presented. This data could be added to a table if it is extensive or, if there are only a few results to present, they can be embedded in the text.
The text in this part of the results section has been clarified and ANOVA details given: “There was statistically significant fat loss after continuous tube feeding compared to the bolus feedings as determined by one-way ANOVA (F(6,56)=7,330, p=.000) with post-hoc Tukey Test. However, no significant difference in fat loss was seen between the horizontal and tilted modalities (p= .875 – 1.000 depending on modality).”.
Line 224: the claim that “mid-IR transmission spectroscopy is being used in the NICUs around the world” needs a reference to support it.
- We have added a reference (the new [19]).
Line 225: really is a typo of rely
- Thank you, Reviewer 2. The typo has been corrected.